# The off-prescription use of modafinil: An online survey of perceived risks and benefits

**Rachel D. Teodorini**[ID]*[◉], **Nicola Rycroft**[◉], **James H. Smith-Spark**[ID][◉]

Division of Psychology, London South Bank University, London, United Kingdom

◉ These authors contributed equally to this work.
* teodorr2@lsbu.ac.uk

**Data Availability Statement:** All relevant data are within the paper and its Supporting Information files.

**Funding:** R.D. Teodorini received a fee-waiver studentship for her doctoral studies from the Centre for Addictive Behaviours Research, London

## Abstract

Cognitive enhancing drugs are claimed to improve cognitive functions such as learning and attention. However, little is known presently about the characteristics of off-prescription cognitive enhancing drug users or their perceived everyday experience with these drugs. As modafinil is the most commonly used off-prescription cognitive enhancing drug, the current study aimed to provide a detailed profile of modafinil users and their experiences and perceptions of this drug. To this end, an online survey, targeting cognitive enhancing drug users and students, was advertised on forum sites. Information was obtained regarding demographic data, illicit drug use, psychiatric diagnosis and experience of modafinil. Of the 404 respondents, 219 reported taking modafinil. Of these the majority were male, American or British, university-educated and currently employed, with a mean age of 27. Overall, modafinil was perceived by users as being safe. Modafinil users reported higher levels of illicit drug use and psychiatric diagnosis than would be expected from population-based data. More frequent reported modafinil use was associated with higher numbers of perceived benefits whilst reported frequency of use was not associated with the number of perceived risks. There was also a tentative link between the reported use of modafinil and the reported presence of psychiatric disorders, largely depression and anxiety. Respondents who had reported a psychiatric diagnosis declared higher subjective benefits of modafinil. This may suggest further beneficial effects of modafinil or it may reflect insufficient medical treatment for psychiatric disorders in some people. Overall, the findings of the current study should be beneficial in informing clinicians and legislative bodies about the modafinil user profile and how modafinil is perceived.

## Introduction

Cognitive enhancing drugs (CEDs) are believed to improve cognitive functions such as attention and motivation [1,2]. They are prescribed for conditions such as dementia, attention deficit hyperactivity disorder (ADHD) and narcolepsy [3, 1]. Further to their prescribed use, off-prescription use of CEDs has been reported, particularly by students during university assessments [4, 5]. Survey data have identified a number of subjective benefits experienced as a result of taking CEDs, such as improved concentration, the ability to study for longer [6], increased alertness [7], increased focus, productivity and drive [8], increased mental stamina or

South Bank University. No other funding was received. The funders had no role in the study design, data collection and analysis, decision to publish or preparation of this manuscript.

**Competing interests:** The authors have declared that no competing interests exist.

endurance [9] and a greater interest in work [10]. The reasons for taking CEDs have been found to include the fear of academic failure, the need to meet high work demands, overcoming procrastination and boosting motivation [11, 12]. Beyond these reasons, some individuals may also be self-medicating to treat undiagnosed attention deficit problems that they are experiencing [6, 12]. Although the perceived effects of CEDs in everyday settings have been investigated [13–15], the CEDs included in such surveys were either looked at more generally as prescription stimulants or the use of a range of different CEDs was surveyed. Given that modafinil is recognised as the most commonly used CED off-prescription [16–18], the research reported in the current paper focused specifically on understanding the user profile and perceived effects of modafinil when taken for non-medicinal purposes.

Modafinil is a mild psychostimulant drug prescribed for narcolepsy [19], sleep apnoea, shift worker sleep disorder [20, 21] and ADHD [22]. The recommended dose for modafinil is 200mg taken once daily [23]. Modafinil has been found to be well-tolerated, with a low incidence of adverse effects and low potential for abuse [24]. It is well absorbed, reaching peak plasma concentration between two and four hours following oral administration, and has a half-life of approximately 12–15 hours [25, 26]. Modafinil's mechanism of action, while not yet fully understood, is complex. It is thought to act primarily through noradrenaline (NE) and dopamine (DA) transporter inhibition [27, 28]. It also acts on serotonin, histamine, gamma-aminobutyric acid and glutamate [29]. It is believed that modafinil's action on orexin also results in increases in the hypothalamic release of histamine [30] and one of the actions of histamine is arousal and wakefulness. Therefore modafinil, when taken at the end of the day or in the evening, may result in an extended period of wakefulness which some individuals may find advantageous, particularly when working towards a pressing deadline. The most common adverse effects of modafinil are headaches, nausea, nervousness, rhinitis, diarrhoea, anxiety and insomnia [27]. In rare cases high doses of modafinil may also induce psychosis [31]. Modafinil has been found to enhance some aspects of cognitive performance in the laboratory. Gilleen et al. [32] administered a 200mg daily dose of modafinil to healthy volunteers over a 10 day period alongside cognitive training. They found that performance on a language learning task, which drew upon attentional, comprehension and working memory processes, was significantly greater in the modafinil group compared with controls. Whilst other studies (e.g., [33]), have reported similar effects of modafinil in non-sleep deprived, healthy individuals, these effects may be stronger when baseline performance is lower [34]. Differences in baseline performance may also explain the results of Repantis et al.'s [18] systematic review of the effects of modafinil in healthy subjects. They found a moderate positive effect for a single dose administration of modafinil on reaction time, divided, sustained and selective attention. However, Battleday and Brem [35] suggested that some of the cognitive tests employed may not have been sensitive enough to detect improvements in healthy, non-sleep deprived adults. Repantis et al. [18] acknowledged that the cognitively enhancing effects of modafinil are greater in sleep-deprived individuals and that the effects of CEDs depend, to a certain extent, on an individual's baseline performance. Randall, Shneerson and File's [36] analysis of the effects of modafinil in healthy students revealed that modafinil only benefitted performance in those with lower IQ, where significant improvements were found in sustained attention, speed of response and visuospatial and constructional ability. It may be that people choose to take modafinil when perceived cognitive demands are high or when their performance may be lowered through some form of impairment such as low baseline levels (i.e. [18]), lower IQ (i.e. [36]) or through the use of other drugs. In unimpaired individuals, modafinil may have little or no enhancing effects, although the basic testing paradigms used in some laboratory-based studies may not be robust enough to detect modafinil's effects [35].

However, the studies reviewed so far have demonstrated the effects of modafinil when measured under laboratory conditions rather than when it is used illicitly, off-prescription, in daily life. Laboratory-based research takes place in controlled environments, providing very useful behavioural, pharmacodynamic and pharmacokinetic information about modafinil use. Nevertheless, off-prescription use of modafinil occurs in uncontrolled environments, often concurrently with other drug use. Drug interactions, dosage levels, perceived effectiveness, motivation and frequency of use and the quality of drugs are all factors which are likely to reflect the real-life experience of modafinil use. Online surveys, therefore, provide important information regarding everyday experience with modafinil that cannot be obtained via laboratory-based research.

Gaining such information is important since, in recent years, health concerns have been raised regarding the off-prescription use of modafinil for cognitive enhancement [37, 2]. Further to these concerns, it appears that modafinil is commonly used for academic study, at least in elite universities in the UK Higher Education sector. For example, one in five students at Oxford University reported the use of modafinil [5] and one in ten students at Cambridge University reported the use of either modafinil, Adderall or Ritalin for the purposes of cognitive enhancement [38].

Despite these health concerns and reported prevalence rates, there is no published study focused on both the positive and the negative perceived effects of modafinil and how these may be related to patterns of use, use of other drugs and psychiatric diagnosis. Therefore, the survey reported in the current paper sought to address this gap in the literature. The main aim of the study was to investigate the modafinil users' perceived experiences of the drug and how this related to frequency of use. A further aim was to gain a greater understanding of the modafinil user's profile via the collection of demographic information, motivations for using modafinil, how they accessed the drug and to what extent they were aware of the dangers of unsupervised use, such as those relating, for example, to dosage levels and dependency.

Strong associations have been found between CED use and the use of illicit drugs such as cannabis, cocaine, amphetamines and MDMA/ecstasy [39–42]. These associations beg the question as to whether illicit drug users are more likely to take CEDs because they are more open to using drugs in general. Therefore, the current study also aimed to investigate concurrent illicit drug use by modafinil users. Modafinil has been found to have mood enhancing effects [43, 44] and has been identified as having therapeutic potential for depression and cocaine dependency [27]. The perceived effects of modafinil amongst drug users and people with a psychiatric diagnosis have not, however, been explicitly investigated outside of the laboratory. Thus, a further aim was to investigate the psychiatric status of modafinil users.

Given that modafinil appears to be used most commonly by students [5, 38], when underperformance is likely [18] or work demands are high [11], it appears that modafinil provides benefits at cognitively demanding times. It seems to offer many potential benefits such as improved attention, speed of response and visuospatial ability [18, 36], yet it also has low incidence of adverse effects [28]. Bearing these points in mind, it was hypothesized that 1) more frequent use of modafinil would yield greater perceived benefits and 2) the perceived benefits would outweigh the perceived negative effects (or risks). Additionally, as it appears that modafinil has the effect of ameliorating poor performance, it seemed very plausible to assume that the benefits provided by modafinil cease to be present once the drug has worn off. Therefore, it was also hypothesized that 3) these reported benefits would not persist beyond the immediate use of modafinil. Finally, bearing in mind the mood-enhancing effects of modafinil [43, 44], it was also hypothesised that 4) individuals with a self-declared psychiatric diagnosis would perceive greater benefits of modafinil use compared with those not reporting a psychiatric diagnosis.

## Method

### Respondents

This study has been ethically approved by the London South Bank University Research Ethics Committee, UREC 1626. Consent was obtained via a consent form found at the start of the online questionnaire. Respondents were only able to proceed with the survey if they clicked on each statement of the consent form. A convenience sample was recruited through online forums (see S1 File in the supplementary materials for the advertisement). Bluelight (http://www.bluelight.org) and Drugs-Forum (http://www.drugs-forum.com) were selected as they are platforms for a wide range of drug users. Reddit (http://www.reddit.com) was selected as it offers specific platforms (sub-Reddits) for discussions of illicit drug and CED use. Members of many of these forums are recreational drug users (and, in some cases, specifically CED users). They, therefore, tend to be well informed about the drugs in question. These forum members, thus, reflect specific populations of drug users. Although seven of the selected sub-Reddits were drug-related, a further four of the selected sub-Reddits were student forums (see S2 File in the supplementary materials for the list of sub-Reddits). As there have been many reports of student use of CEDs during assessment periods [3, 4], the Student Room (http://www.thestudentroom.co.uk) and the student forum sub-Reddits were selected in order to obtain a broader picture of modafinil use than could be obtained through drug user forums alone. Due to the anonymous nature of data collection, however, it was not possible to separate the data collected from these two population samples. The respondents were recruited by posting an advertisement with a link to the survey on all of the forum sites. No reward was offered for their participation.

A total of 404 respondents completed the survey, of whom 117 reported no use of modafinil and 68 reported prescribed (i.e. medical) use. As this study focused on individuals who reported using modafinil and reported choosing to do so specifically for the purposes of cognitive enhancement, the data from both of these types of respondents were removed prior to statistical analysis. Excluding the data obtained from these respondents resulted in a final sample size of 219. Of the sample, 46.1% were in full-time employment, 26.5% were in part-time employment (both paid and unpaid) and 27.4% were unemployed. However, due to the way in which the question was constructed, it was not possible to determine whether any of the unemployed were students.

### Materials

The survey was constructed following an analysis of other recent drug and CED-user surveys [45–47] and the identification of outstanding questions from the literature regarding motivations for use and access to CEDs [2]. Qualtrics<sup>XM</sup> survey software was used to create the online survey (see S3 File in the supplementary materials for the full questionnaire).

After the presentation of an information sheet and consent form, the questionnaire was presented and ended with a debriefing. A maximum of 51 questions were asked, the total number varied according participants' responses, and, as a result, the time to complete the questionnaire ranged between approximately five and 25 minutes. The individual sections of the questionnaire were as follows:

**Demographics.** In order to gain a greater insight into the profile of the modafinil user frequenting these forum sites, demographic information was collected. This section consisted of nine questions, covering age, gender, nationality, educational and employment details.

**Psychiatric health and drug use.** This section included 22 questions relating to psychiatric diagnosis, psychiatric treatment and drug use history (cannabis, cocaine, amphetamines

such as speed, and MDMA/ecstasy). Example questions were "Have you ever been diagnosed with a psychiatric condition?" and "What was the diagnosis?". The term 'psychiatric diagnosis' was used in order to highlight that a formal diagnosis of mental health issues was required and to avoid any cultural or international differences in how the term 'mental health' might have been perceived by respondents. Given that the questions were framed in this way, the term 'psychiatric diagnosis' has continued to be used when reporting the responses to these questions in the Results section. Drug use questions required respondents to indicate, with a yes or no response, lifetime use, use in the last year and attendance at drug and alcohol treatment programmes (e.g. "Have you ever been treated for a drug or alcohol-related problem?").

**Modafinil use.** This section included 16 questions on frequency of use, dosage taken, how modafinil was obtained, concurrent use of other drugs, motivations for use, and perceived risks and benefits experienced after taking modafinil. A full list of respondents' reported concurrent use of other drugs can be found in the S1 Table of the supplementary data.

Questions regarding modafinil were presented with the brand names Provigil and Modalert to ensure it was not confused with armodafinil which, although similar to modafinil, is a different drug and may have different effects. Armodafinil is the *R*-isomer of racemic modafinil and has been shown to result in higher plasma concentrations of the drug late in the day and so may result in prolonged wakefulness [29]. As studies investigating armodafinil have focused primarily on its waking effect, there is a lack of literature focused on its effects on cognition and, to the authors' knowledge, no study has compared the cognitive enhancing effects of armodafinil and modafinil.

A list of 14 known positive effects (or benefits) of modafinil (which included "none") was presented, for example, "increased concentration", "motivation", "clarity of mind", "ability to focus" and "alertness". The list was chosen rather than providing a free text response in order to avoid respondents misinterpreting the nature of what was required by the question and to facilitate data entry and analysis. This list was drawn from previous surveys and reviews of modafinil and other CEDs [18, 48, 49] and from online forum posts on the Reddit site (http://www.reddit.com/). A list of known negative effects (or risks) of modafinil was drawn from the pharmaceutical data sheet for modafinil [23]. A total of 24 negative effects (including 'none') was also presented. These included for example, "anxiety", "diarrhoea", "headache", dry mouth" and "insomnia".

Both lists were presented twice, with respondents asked to self-report their experiences with modafinil during two timeframes, namely 'immediate–whilst on the drug' and 'longer-lasting–once the drug has worn off'. All four lists had the option for respondents to select as many items as they felt applied to them, as well as the option to tick 'other' which, if selected, brought the participant to a text box where further effects could be added via keystrokes. A full list of the benefits and risks presented to the respondents, together with their free text data beyond the listed items, can be found in the S2 Table of the supplementary materials.

Two further questions about modafinil were included to assess knowledge of recommended dosages, perceptions of harmful use and dependency on modafinil. These were "How much modafinil do you think it is safe to take at any one time?" (with a single response to be chosen from the following options: none, 50mg, 100mg, 200mg, up to 400mg, and more than 400mg), and "Do you feel dependent on modafinil?" (requiring a yes or no response).

## Analysis

A mixed-measures design was used to investigate the perceived effects of modafinil and its frequency of use. There was one between-group factor, the frequency of use of modafinil (with five levels: every day, three or more times a week, once or twice a week, two or three times a

month and six times or less a year). The two within-group factors were the timeframe over which the effects of modafinil were reported (with two levels: immediate and longer-lasting) and the perceived effects of modafinil (with two levels: benefits and risks). A between-groups design was used to investigate whether reported psychiatric diagnosis status, the dependent variable, had an influence on perceived effects of modafinil, the independent variable.

Data relating to frequency of use of modafinil were analysed using SPSS software, version 21. A three-way mixed-measures analysis of variance (ANOVA) was conducted. Five x 2 x 2 ANOVAs were used to explore the differences between the perceived positive and negative effects of modafinil and how these related to frequency of use. As the highest number of positive effects selected by any respondent was 14 (n = 1) and the highest number of negative effects selected was 7 (n = 1) (not including options of 'none' and 'other'), these data were treated as continuous. The large number of cells with uneven cell sizes would otherwise have prevented a coherent analysis of categorical data. Bonferroni post-hoc tests were used to determine significant differences in the number of effects reported for different frequencies of use. To facilitate the interpretation of a 2 x 5 interaction, post-hoc t-tests were used. Furthermore, Cohen's d was calculated using the mean difference between the groups and dividing this by the pooled standard deviation to determine the size of the difference between positive and negative effects in each frequency of use group.

Mann-Whiney U tests were performed to establish whether psychiatric diagnosis status had any impact on perceived effects of modafinil. For this analysis, the total number of perceived risks was subtracted from the total number of perceived benefits. This created a risk-benefit trade-off value with positive scores indicating that the perceived benefits outweighed the perceived risks. These scores were calculated to enable comparisons to be made between those with and without a reported psychiatric diagnosis.

## Procedure

A link to the survey, along with an advertisement, was posted, with appropriate permission, to the selected forum sites. The survey was conducted over a two-month period from August 12th to October 12th, 2016. Respondents were asked to confirm they were aged over 18 years and not under the influence of a psychoactive drug whilst completing the survey. Once the consent form had been clicked to indicate consent, the survey commenced. It ended with a debriefing text which required the selection of a submit button for the data to be logged and included in the analyses.

## Results

### Demographics

The survey respondents reported themselves to be predominantly male (86%, N = 188) and aged 18–68 years with a mean age of 27 years (SD = 9.85). Over half the respondents were either American (36%, N = 73) or British (27%, N = 54), with European (11%, N = 22), Australian (9%, N = 19), Canadian (6%, N = 13) and 'Other' (11%, N = 21) nationalities also being reported. Less than 1% of the sample indicated that they were educated up to the age of 16 (N = 1) and 37% (N = 80) reported that they were educated up to the age of 18. The remainder reported that they held undergraduate (43%, N = 93) or postgraduate (21%, N = 45) degrees. A total of 54% (N = 118) of respondents reported that they were currently studying for a qualification and 43% (N = 95) reported that they were university students. Therefore, the majority of respondents currently studying (reportedly) said they were university students (80%, N = 95).

### Psychiatric diagnosis

The proportion of respondents who reported a psychiatric diagnosis was 22% (N = 47). The most commonly reported diagnoses were 'Depression' (10%, N = 21), 'Anxiety' (1%, N = 3), 'Depression with Anxiety' (6%, N = 13) and 'Other' (5%, N = 10).

### Illicit drug use

Cannabis was the most commonly reported illicit drug used by the respondents (lifetime use, 83%, N = 180, past year use, 60%, N = 112), followed by MDMA (lifetime use, 47%, N = 103, 47%, past year use, 29%, N = 49), cocaine (lifetime use, 41%, N = 89, past year use 21%, N = 35) and amphetamines (lifetime use, 46%, N = 98, past year use, 26%, N = 45).

### Access to modafinil

The respondents were asked to indicate how they obtained modafinil, selecting more than one option if applicable. The most commonly reported means of access was via online sources (78%, N = 170), followed by access via a friend (8%, N = 18), a dealer (7%, N = 16), and someone else's prescription (2%, N = 4). If modafinil was obtained via sources not presented in the survey, the respondents selected 'other' (10%, N = 22) which covered access (such as 'over-the-counter') in a country where it is legal to purchase the drug.

### Motivations for use of modafinil

The most commonly reported motivation for modafinil use was to improve attention/focus (84%, N = 183). The remaining reported motivations for use were 'to work long hours' (54%, N = 119), 'to get more done' (78%, N = 169), 'exams' (33%, N = 71), 'night work' (19%, N = 42), 'to think more clearly' (55%, N = 120), and 'other' (13%, N = 28).

### Frequency of modafinil use and perceived risks and benefits of modafinil

The data analyses reported in this subsection are based on the total number of boxes ticked in the perceived risks and benefits section of the questionnaire. Respondents reporting that they took modafinil on a daily basis reported the greatest number of perceived effects (benefits and risks) across both timeframes (immediate and longer lasting). Means, as well as the minimum and maximum numbers of total effects, are shown in Table 1.

There was a significant main effect of frequency of modafinil use on the number of effects reported, ($F_{(4, 214)} = 6.91$, $MSE = 7.42$, $p < .001$, $\eta_p^2 = .114$). Compared with the respondents whose reported modafinil usage was six or fewer times per year, Bonferroni post-hoc tests confirmed significant differences in the number of effects reported between those who reported taking modafinil once or twice per week ($p = .010$), three times or more per week ($p = .007$), and every day ($p < .001$). A significant difference was also found in the number of effects reported between those respondents who reported taking modafinil two or three times per month and those who reported taking it every day ($p = .006$).

The respondents reported experiencing more immediate effects (mean = 4.99, $SE = 0.14$) than longer-lasting effects (mean = 2.07, $SE = 0.10$) and there was a significant main effect of timeframe of modafinil use, ($F_{(1, 214)} = 465.21$, $MSE = 3.648$, $p < .001$, $\eta_p^2 = .685$).

The respondents reported more benefits (mean = 5.26, $SE = 0.17$) than risks (mean = 1.80, $SE = 0.07$). There was a significant main effect of perceived effects on modafinil use, ($F_{(4, 214)} = 379.3$, $MSE = 6.264$, $p < .001$, $\eta_p^2 = .639$).

There was a significant interaction between timeframe and frequency of use of modafinil, ($F_{(4, 214)} = 2.53$, $MSE = 3.648$, $p = .041$, $\eta_p^2 = .045$), see Fig 1. Post-hoc within-subjects t-tests

**Table 1. Frequency of modafinil use and means for both benefits and risks of modafinil.**

| Reported frequency of modafinil use | N (%)* | Mean (SD) overall no. of effects | Min no. of overall effects** | Max no. of overall effects** |
|---|---|---|---|---|
| Every day | 26 (11.90) | 4.39 (0.27) | 1.00 | 8.00 |
| Three or more days/ week | 66 (30.10) | 3.68 (0.17) | 1.25 | 8.5 |
| Once or twice/week | 52 (23.70) | 3.70 (0.19) | 1.25 | 7.00 |
| Two or three times/month | 38 (17.40) | 3.17 (0.22) | 1.00 | 6.50 |
| Six times or fewer per year | 37 (16.90) | 2.27 (0.22) | 1.00 | 4.75 |

Respondents (N = 219)

*Percentages relate to the number of respondents within each frequency of use category.

**Effects are collapsed across timeframe and perceived effects, therefore the minimum and maximum numbers reported in the table may not reflect whole numbers.

confirmed that the difference between immediate and longer lasting effects was significant for all five frequency of use groups (see S3 Table of the supplementary material for further information). Data presented in the 2 x 2 x 5 ANOVAs were not normally distributed. The results of the ANOVAs run on log transformed data are available in the supplementary materials (S4 Table). The overall pattern of the results remained the same after log transformation.

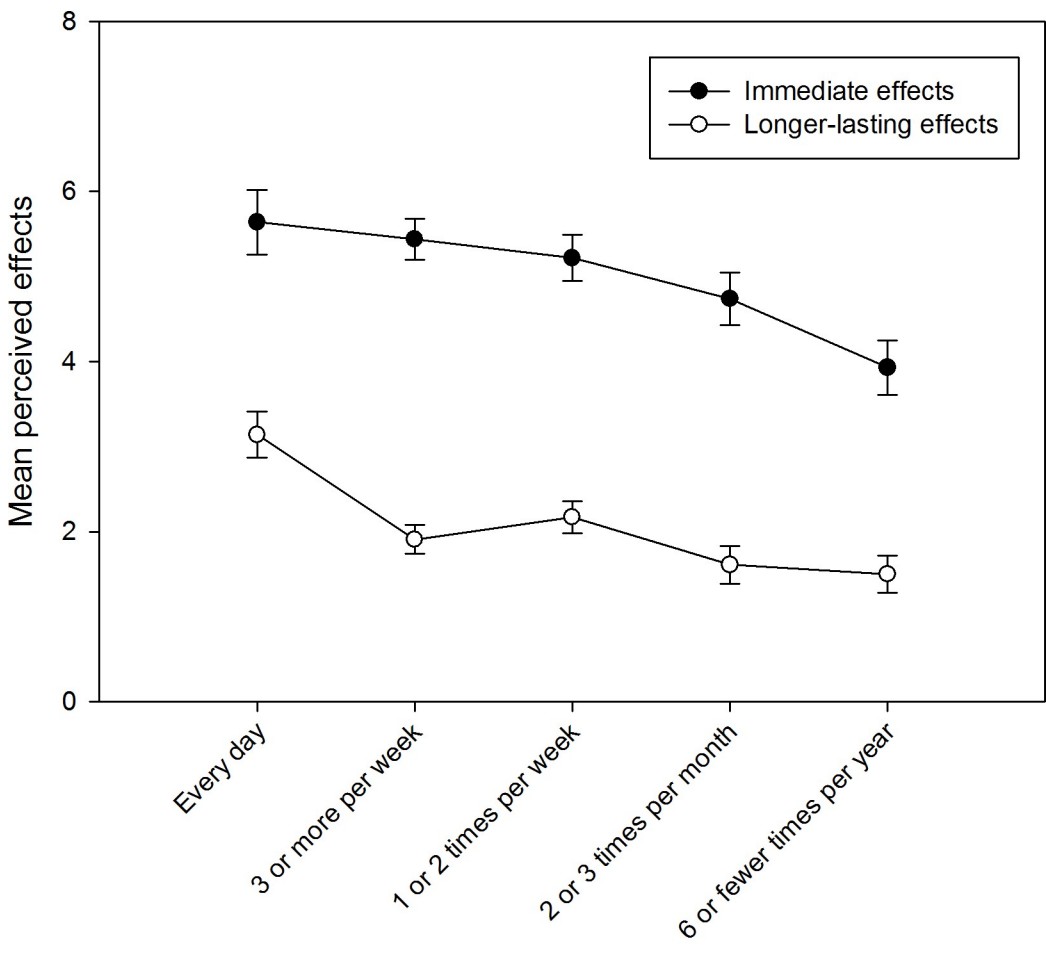

**Fig 1. Interaction between timeframe of perceived effects and frequency of use of modafinil.**

In order to determine the relative magnitude of the differences between the perceived immediate and longer-lasting effects, Cohen's *d* was calculated to establish the effect size for the difference for each frequency of use group. The results indicated that the effect size was smaller for both everyday use and six or fewer times a year than for all other frequency of use groups. The two-way interaction appears, therefore, to be due to a smaller difference between the reported immediate and longer-lasting effects in the most and least frequent use groups. Fig 1 suggests that every day users reported a higher number of long-term effects (risks and benefits combined) and those that reported a frequency of use of six times or fewer reported fewer immediate effects (both risks and benefits combined).

There was a significant interaction between perceived effects and the reported frequency of use of modafinil, ($F_{(4, 214)} = 4.597$, $MSE = 6.264$, $p < .001$, $\eta_p^2 = .079$), and this is plotted Fig 2. Post-hoc within-subjects t-tests confirmed that the difference between perceived benefits and risks was significant for all five frequency of use groups (see S5 Table of the supplementary material for further information). Cohen's *d* was calculated to establish the effect size for the difference between risks and benefits for each frequency of use group. The effect size was smallest for six or fewer times a year. Fig 2 shows that the reported frequency of use of modafinil of less than once a month yielded a smaller difference between its perceived risks and benefits.

Further to this, the interaction between timeframe and perceived effects was also significant, ($F_{(1, 214)} = 313.32$, $MSE = 2.739$, $p < .001$, $\eta_p^2 = .594$). The immediate benefits were higher than the immediate risks (mean$_{(benefit)}$ = 7.77, SE = 0.24, mean$_{(risk)}$ = 2.22, SE = 0.11, $t = 22.758$, df = 218, $p < .001$, $d = 1.54$). Longer-lasting benefits were also higher than longer-lasting risks (mean$_{(benefit)}$ = 2.76, SE = 0.17, mean$_{(risk)}$ = 1.38, SE = 0.06, $t = 8.111$, df = 218, $p < .001$). The effect size for the difference between immediate risks and benefits was higher ($d = 1.54$) than the effect size for longer-lasting risks and benefits ($d = 0.54$).

The three-way interaction between perceived effects, timeframe and frequency of use of modafinil was not statistically significant, ($F_{(4, 214)} = 1.53$, $MSE = 2.739$, $p = .195$).

## Dosage and dependency

Half the respondents (N = 111, 51%) reported that they considered a dose of 200mg of modafinil to be safe to take at any one time, 31.5% (N = 69) felt that a dose of up to 400mg was safe, 11% (N = 23) reported feeling that a dose of more than 400mg was safe, 6% (N = 12) felt that a dose of 100mg was safe and 2% (N = 4) reported feeling that 50mg was a safe dose.

Only 6% (N = 12) of respondents reported feeling dependent on modafinil, all of whom reported taking modafinil at least once or twice a week.

## Effects of psychiatric diagnosis on perceived effects of modafinil

The number of the immediate effects of modafinil which were reported differed based on the respondents' reported psychiatric diagnosis status. Individuals who reported a psychiatric diagnosis reported experiencing greater longer-lasting benefits of modafinil than those without a diagnosis (see Table 2).

## Discussion

The overall aim of the survey was to investigate the modafinil users' perceived experiences of the drug and how this related to frequency of use. The results indicate that modafinil was perceived as having greater benefits than risks and a greater reported frequency of use was found to result in greater reported benefits. The majority of respondents reported themselves to be male, employed and university-educated. The perceived dependency on modafinil was low,

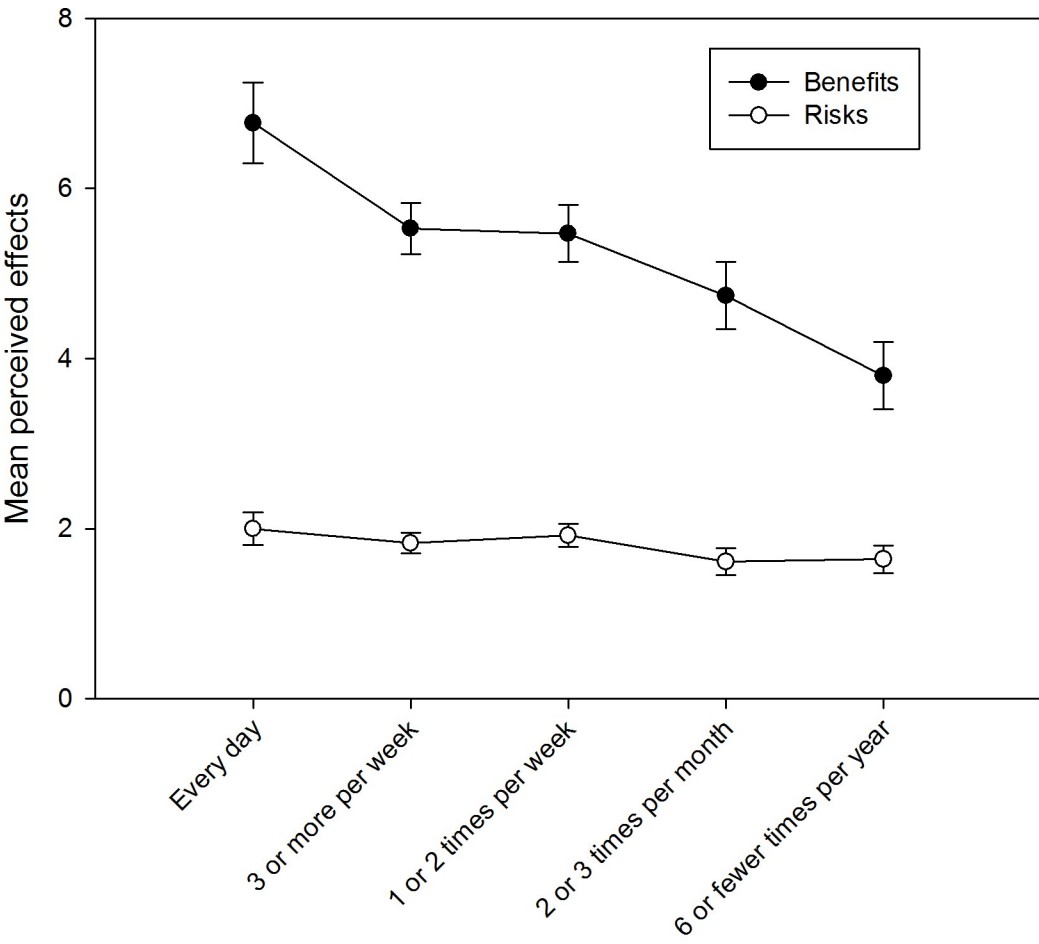

**Fig 2. The interaction between the number of perceived effects and the frequency of use of modafinil.**

despite 12% of the sample using modafinil every day. Overall, modafinil was seen as being a safe drug, even when taken three times a week or more. Most respondents appeared to be aware of the recommended dose of 200mg. Although perceived dependency on modafinil was low, the link between frequency of use and perceived benefits suggest that there is a possibility that dependency may develop over time.

**Table 2. Effects of mental health diagnosis on perceived effects of modafinil.**

|  | Yes/No*** (N) | Immediate effects* | | | Longer-lasting effects* | | |
|---|---|---|---|---|---|---|---|
|  |  | **Mean (SD)** | **M-W**** | ***p*** | **Mean (SD)** | **M-W**** | ***p*** |
| **Psychiatric diagnosis** | Yes (48) | 5.10 (4.01) | 3737.5 | 0.343 | 1.98 (2.88) | 3259.0 | 0.021 |
|  | No (171) | 5.65 (3.48) |  |  | 1.16 (2.28) |  |  |

Respondents (N = 219)

*Scores reported are a 'risk-benefit' trade off calculated by subtracting the number of negative effects from the number of positive effects.

** M-W = Mann-Whitney U.

*** Yes/No indicates those who had not reported having had a psychiatric diagnosis.

The data show that more frequent reported use of modafinil led to perceived benefits. It appears that the reported use of modafinil on at least a monthly basis resulted in a higher number of reported benefits whilst reported risks remain low. A plausible explanation could be that more frequent reported use occurs as a consequence of greater perceived benefits since it seems unlikely that continued use would occur without experiencing the benefits of the drug. Chronic drug use is, however, known to lead to tolerance caused, in part, by a reduction in receptor numbers [50] but this may not always be the case with modafinil. Nasr, Wendt and Steiner [51] reported that long-term use of modafinil in patients with affective disorders did not induce tolerance. Therefore it is plausible that continued, long-term use of modafinil could still provide these perceived benefits.

The data also show that perceived benefits outweigh perceived risks. The respondents reported significantly more benefits than risks. It is known that modafinil is well-tolerated and lacks the undesirable adverse effects of other stimulants [24]. It was, therefore, expected that the benefits would outweigh the risks based on the range of potential benefits that it offers, which include enhanced attention, comprehension and working memory [32], as well as alertness, vigilance and enhanced executive functions [52].

Perceived benefits, however, did persist beyond immediate use of modafinil. Everyday use of modafinil resulted in higher reported longer-lasting effects. While relatively high effect sizes might be expected when exploring differences between immediate and longer-lasting effects of modafinil, it was not expected that there would be a higher reported level of longer-lasting effects in the 'every day' user group. To the authors' knowledge, modafinil has not been found to exhibit any significant positive neuroplastic changes in humans. It was, therefore, expected that modafinil's positive effects would no longer be reported to be experienced when the drug has ceased to be active. This finding may, however, be explained by the pharmacokinetic profile of modafinil. As the half-life of modafinil is approximately 12–15 hours [25, 26], daily use of modafinil would result in a constant, and possibly increasing, plasma concentration of modafinil, which also suggests constant, higher levels of synaptic DA and NE (in addition to other neurotransmitters that are modulated by modafinil). Increased synaptic DA and NE have been associated with improved cognitive function [28, 53]. It would seem reasonable, therefore, to argue that everyday use would lead to greater reported long-term benefits as levels of modafinil would decline to approximately 25% (i.e. two half-lives) by the time the next dose was taken, resulting in increased concentrations of modafinil in the blood. Modafinil has also been found to have an effect on the glutamate receptors of the hippocampus at ascending doses [29] and may well have a supportive effect on long-term potentiation and positive neuroplastic changes which could also explain the increase in longer-lasting effects in the 'everyday' user group.

One-fifth of respondents reported having a psychiatric diagnosis and, of these, the most commonly reported diagnoses were depression, anxiety or both depression and anxiety. However, it should be noted that the reported rates of psychiatric diagnosis among modafinil users do seem to be higher than the 18.3% (of adults in the USA) stated in the 2016 National (USA) Survey on Drug Use and Mental Health [54]. Prevalence (and official recognition) of psychiatric problems varies by country, gender and age group [55, 56], therefore making direct comparisons difficult. It may be that modafinil use may have preceded the psychiatric diagnosis or it may be that modafinil may have been being used to combat the potential side-effects of prescribed medications.

Individuals with a psychiatric diagnosis did not always perceive greater benefits of modafinil use compared with those without a psychiatric diagnosis. Respondents with a reported psychiatric diagnosis perceived significantly greater longer-lasting benefits but the perceived immediate benefits were not significantly greater. This is somewhat surprising considering the potential mood enhancing effects of modafinil [43]. Deficiencies in DA, NE and serotonin are

likely to underpin major depressive disorder [57]. As modafinil intake results in higher levels of DA, NE, and serotonin [29], it is possible (although highly speculative) that the benefits of these drugs for people with a psychiatric diagnosis may be due to poorer functioning of these neurotransmitter systems. A range of cognitive impairments have been reliably associated with depression [58] and some anxiety disorders [59]. This raises the possibility that modafinil may be used to enhance poorer cognitive performance amongst this population.

As would be expected from recruitment via online drug user forums, levels of reported illicit drug use in the sample were high. Compared with data from the European Monitoring Centre for Drugs and Drug Addiction [60] and the National Institute of Drug Abuse [61], modafinil users reported levels of drug use that were between 20% (for cocaine) and 50% (for cannabis) higher than one would expect to see amongst the general population of North America and Europe. These findings are consistent with those of Ott and Biller-Andorno [62] who reported similar percentages of lifetime illicit drug use to those reported in the current study. Slightly lower rates of illicit drug use amongst Swiss CED users were reported by Maier, Haug and Schaub [40]. The findings in the current study may indicate, as suggested earlier, that illicit drug users are more likely to take CEDs since they report being more open to taking drugs in general. Conversely, although the reported illicit drug use in the current study may reflect the drug user forum population from which the sample was drawn, it does suggest that those who take CEDs tend to use recreational drugs as well.

The present study has a number of limitations. As the current data are based on self-reports, it is possible that the presence of demand characteristics have influenced respondents to provide answers in line with what they perceived as being the aims of the study or, perhaps, it may reflect self-justification of respondents' use of modafinil. Additionally, although the gender bias in the sample may reflect a higher propensity of risk-taking behaviour by men [63], it may more simply be an artefact of the online forum recruitment methods as users of such forum sites appear typically to be male [64–66]. The questionnaire did not include questions on current use of psychiatric medications or other medications which could potentially interact, attenuate or increase the effects of modafinil and other medications. It is difficult, therefore, to assess the benefits of modafinil in those with a psychiatric diagnosis in the absence of this information. Further research is needed to explore whether there is any evidence to indicate that people with a psychiatric diagnosis self-medicating via the off-prescription use of modafinil.

As many respondents were recruited via drug forums, drug-use data from modafinil users are likely to reflect the behaviour of this population sample and may not reflect all modafinil users. However, this population was targeted to gain a greater understanding of how these drugs were experienced and perceived by users themselves. The survey has thus provided important information on the profile of the off-prescription user of modafinil but the data are subject to the usual limitations of online surveys [67, 68], such as a reliance on self-report measures with no check being possible on the identity of the respondent. In addition, this was a self-selecting sample of people who use student, CED user, and drug user online forums. The demographic profile of 'modafinil users' in this study must be interpreted in the light of the demographics of people who use online forums. Whilst these weaknesses must be acknowledged, the current study is comparable with the approach taken in other published studies [45, 68].

A further limitation is that the survey did not include questions testing for attentional problems. This is potentially a concern as it is possible that there may be an element of self-medication by individuals with low baseline levels of DA and NE. Further research is thus warranted to investigate this possibility.

Finally, the survey did not include questions on nicotine and alcohol use, use of other CEDs or routes of administration of modafinil. The authors intend to address these issues in a further study, which will include these questions and will also investigate cognitive and attentional functions both with and without modafinil use.

## Conclusion

To the authors' knowledge, this is the first paper to report a detailed survey into the perceived experience of modafinil and the modafinil user's profile. This study has found that, as the reported frequency of modafinil use increased, the number of perceived benefits increased, whilst the number of negative effects remained stable and unchanged. Respondents also reported significantly more benefits than risks and more immediate benefits than longer-lasting benefits. Conversely, those with a reported psychiatric diagnosis perceived greater longer-lasting benefits compared with those without a psychiatric diagnosis. This study has provided insight into the profile of modafinil users who are, in this English language study at least, mostly male, American or British, educated, employed and in their mid-20s. Overall, modafinil was perceived by users as being safe. There was a pattern of reported recreational drug use associated with modafinil use. There was also a tentative link between the reported use of modafinil and the reported presence of psychiatric disorders, largely depression and anxiety.

This paper has, therefore, highlighted a potential concern over the perception of modafinil as a 'safe' drug, even when taken on a weekly or daily basis. Whilst self-reported dependency was low in this sample, the link between perceived benefits and frequency of use suggests that there may be the potential for dependency to develop over time. The possibility that people may be using modafinil as a supportive treatment for a psychiatric diagnosis warrants further explanation from a public health or clinical use perspective. This link may be suggestive of further beneficial effects of modafinil or, more simply, it may reflect insufficient medical treatment for psychiatric disorders in some people. Whichever explanation turns out to be correct, it is clear from this paper that the reported perceptions of modafinil as a safe drug, with more frequent use giving greater benefits, suggests that off-prescription use may well increase in popularity and this may result in dependency in some people.

## Supporting information

**S1 File. Advertisement.**
(DOCX)

**S2 File. List of subReddit forum sites.**
(DOCX)

**S3 File. Modafinil and methylphenidate questionnaire.**
(DOCX)

**S4 File. Study data.**
(SAV)

**S1 Table. List of concurrent drug use.**
(DOCX)

**S2 Table. Full list of positive and negative effects including free text answers.**
(DOCX)

**S3 Table. Timeframe & frequency of modafinil use post-hoc t-test & Cohen's d.**
(DOCX)

**S4 Table. Log-transformed data for 2 x 5 x 5 ANOVAs.**
(DOCX)

**S5 Table. Perceived benefits and risks & frequency of modafinil use post-hoc t-test & Cohen's d.**
(DOCX)

## Acknowledgments

The authors would like to thank the forum administrators at Reddit, Bluelight and The Student Room who assisted in running this survey, and the participants for their valued contribution to the research.

## Author Contributions

**Conceptualization:** Rachel D. Teodorini, Nicola Rycroft, James H. Smith-Spark.

**Data curation:** Rachel D. Teodorini.

**Formal analysis:** Rachel D. Teodorini, Nicola Rycroft, James H. Smith-Spark.

**Investigation:** Rachel D. Teodorini, Nicola Rycroft, James H. Smith-Spark.

**Methodology:** Rachel D. Teodorini, Nicola Rycroft, James H. Smith-Spark.

**Project administration:** Rachel D. Teodorini.

**Software:** Rachel D. Teodorini, Nicola Rycroft, James H. Smith-Spark.

**Supervision:** Nicola Rycroft, James H. Smith-Spark.

**Validation:** Rachel D. Teodorini, Nicola Rycroft, James H. Smith-Spark.

**Visualization:** Rachel D. Teodorini, Nicola Rycroft, James H. Smith-Spark.

**Writing – original draft:** Rachel D. Teodorini.

**Writing – review & editing:** Rachel D. Teodorini, Nicola Rycroft, James H. Smith-Spark.

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
