## [Decision Letter · Decision Letter 0]

5 Sep 2019

PONE-D-19-19392

The off-prescription use of modafinil: An online survey of perceived risks and benefits

PLOS ONE

Dear Ms Teodorini,

Thank you for submitting your manuscript to PLOS ONE. After careful consideration, we feel that it has merit but does not fully meet PLOS ONE’s publication criteria as it currently stands. Therefore, we invite you to submit a revised version of the manuscript that addresses the points raised during the review process.

The two reviewers addressed several major and minor concerns about your manuscript. Please revise your manuscript carefully.

We would appreciate receiving your revised manuscript by Oct 20 2019 11:59PM. To enhance the reproducibility of your results, we recommend that if applicable you deposit your laboratory protocols in protocols.io, where a protocol can be assigned its own identifier (DOI) such that it can be cited independently in the future. For instructions see: http://journals.plos.org/plosone/s/submission-guidelines#loc-laboratory-protocols

We look forward to receiving your revised manuscript.

Kind regards,

Kenji Hashimoto, PhD

Academic Editor

PLOS ONE

Journal Requirements:

1. Please include additional information regarding the survey or questionnaire used in the study and ensure that you have provided sufficient details that others could replicate the analyses. For instance, if you developed a questionnaire as part of this study and it is not under a copyright more restrictive than CC-BY, please include a copy, in both the original language and English, as Supporting Information.

Reviewers' comments:

Reviewer's Responses to Questions

**Comments to the Author**

1. Is the manuscript technically sound, and do the data support the conclusions?

Reviewer #1: Partly

Reviewer #2: Yes

2. Has the statistical analysis been performed appropriately and rigorously? 

Reviewer #1: No

Reviewer #2: Yes

3. Have the authors made all data underlying the findings in their manuscript fully available?

Reviewer #1: Yes

Reviewer #2: Yes

4. Is the manuscript presented in an intelligible fashion and written in standard English?

Reviewer #1: Yes

Reviewer #2: Yes

5. Review Comments to the Author

Reviewer #1: General Comments

The authors have presented interesting descriptive data regarding the use of modafinil as a cognitive enhancing medication. The background information given in the introduction was well written and clear.

The aims and hypotheses seem unclear throughout, mostly in the results section, where a series of results is given without much explanation in the introduction or methods about the purpose of these analyses. Apart from the descriptive statistics, the statistical analysis in this paper is unclear. A thorough review of the statistics is required by the authors, making sure that appropriate statistical tests are used for different types of data.

Specific Comments

Introduction

- Some of the references for modafinil are outdated – and reference 6 seems incorrect

- Does the study consider use of armodafinil or just modafinil? Armodafinil is the r-enantiomer of racemic modafinil and often the two are used interchangeably or quite similarly. Can the authors comment on this?

- The start of the final paragraph on page 7 is repetitive

- Were the aims and hypotheses made a-priori or post hoc? Was there a primary hypothesis?

Methods

- The design section describes the statistical methods – I believe this section should report the study design, not stats used.

- Why did the authors exclude those who were prescribed modafinil? They could have been prescribed it and still be using it for cognitive enhancement. Additionally, why did people complete the survey if they were not modafinil users – could they have been past users? I would like to see some information about these people who were excluded as it may be of interest

- The Analysis section requires further description of the statistical tests performed

- Under “Procedure” the specific ethics committee details and approval number should be given

Results

- In Table 1 the authors have presented both benefits and risks together. I think it would be of more interest to separate the effects into the two as it may be different and of interest to readers to present it that way? Can the authors justify their hypothesis here? The hypothesis described in the introduction does not match the presentation of the results here.

- The statistics used to describe the effects of frequency of use and number of effects are unclear. This is ultimately categorical data which seems to have been initially represented like it is continuous, then a series of comparisons have been made and p values given, but no comparative statistic given. The statistical methods should be better described in the methods and should be clearly reported.

- At the end of page 15 there are a series of values given as plurals. Perhaps this information would be better placed in a table in the supplementary material if need be.

- It is unclear how the Cohen’s d values listed at the top of page 16 were calculated. Again this appears to be count data, and should not really be converted to mean values. The Cohen’s d values reported reflect very large effect sizes, but this is not commented on further.

- Figures should be used to support the presentation of data to support the main findings. Neither of the figures match hypotheses outlined in the introduction.

Discussion

- The initial paragraph of the discussion should describe the main results and then start to place them in context of what is known. The discussion of limitations should be saved until later in the discussion section.

- Again the primary and secondary hypotheses should be outlined clearly here with reference to the results.

- The authors acknowledge that selection of a sample from groups of people who discuss the use of the drug as a cognitive enhancer online may be different to normal users. Do we know how many people use the drug as a cognitive enhancer?

Reviewer #2: This is an interesting study on the off-prescription use of modafinil. The data presented here would provide valuable information for considering the pros and cons of modafinil as a cognitive enhancing drug (CED).

The paper is well written in general, but a few modifications would help readers to understand the significance of the data more straightforwardly. Please consider the following points:

1) Introduction

This section is a little bit too long and the specific topics of modafinil and CEDs in general are mixed together.

For example, after the introduction on modafinil, concomitant use of CEDs (general) and illicit drugs appeared (p. 4, paragraph 2, ‘Strong associations…’). Then the story goes on to the effects of modafinil in laboratory settings. After that, problems about studies on CEDs in general appear again (p. 5, the last paragraph, ‘However, these studies…’ and subsequent paragraphs, p. 6. ‘Survey data…’, ‘Although the perceiver effects of CEDs’).

I would like authors to emphasize more straightforwardly (1) why you focused on modafinil, and (2) why you employed an online survey.

2) Results

2-1) Could you show several typical examples of perceived risks and benefits? This would help readers to determine what kind of benefits and risks the users perceived.

2-2) A total of 15 positive effects and 24 negative effects were listed (p. 11) but the reported numbers of overall effects (both positive and negative) ranged from 2.27 to 4.39 (Table 1). Does this mean the majority of users did not perceive any effects? Am I right? If so, please mention this briefly. Plus, the rationale for combining both positive and negative effects should be explained.

3) This section is also a little bit too long. To organize this section better, please consider the following points:

3-1) Considerations on the biochemical mechanisms of action of modafinil and their relevance to motivation for use (p. 21, paragraphs 3 and 4, ‘Deficiencies in DA, NE…’, ‘As reported by Cools et al…’) are over-speculation and should be more concise. Plus, please consider referring to action on the histaminergic system. This seems important for night-time use of modafinil.

3-2) As the authors mentioned, this study has several limitations, such as gender bias, source of information from drug forums, and lack of questions on attentional problems. In this manuscript, these are scattered in several paragraphs. I think it is better to create a new paragraph and compile these topics. Plus, in my opinion, several important questions were missing, such as status of cigarette smoking, use of other CEDs, and perceived levels of academic achievement. Could you include these limitations?

6. PLOS authors have the option to publish the peer review history of their article (what does this mean?). If published, this will include your full peer review and any attached files.

Reviewer #1: No

Reviewer #2: Yes: Naoyuki Hironaka

---

## [Author Response · Author response to Decision Letter 0]

16 Oct 2019

Please see attached document 'Response to Reviewers'

---

## [Decision Letter · Decision Letter 1]

1 Nov 2019

PONE-D-19-19392R1

The off-prescription use of modafinil: An online survey of perceived risks and benefits

PLOS ONE

Dear Ms Teodorini,

Thank you for submitting your manuscript to PLOS ONE. After careful consideration, we feel that it has merit but does not fully meet PLOS ONE’s publication criteria as it currently stands. Therefore, we invite you to submit a revised version of the manuscript that addresses the points raised during the review process.

The reviewer #1 still have some major concerns about the analysis methods implemented in this study. Please revise your manuscript carefully.

We would appreciate receiving your revised manuscript by Dec 16 2019 11:59PM. To enhance the reproducibility of your results, we recommend that if applicable you deposit your laboratory protocols in protocols.io, where a protocol can be assigned its own identifier (DOI) such that it can be cited independently in the future. For instructions see: http://journals.plos.org/plosone/s/submission-guidelines#loc-laboratory-protocols

We look forward to receiving your revised manuscript.

Kind regards,

Kenji Hashimoto, PhD

Academic Editor

PLOS ONE

Reviewers' comments:

Reviewer's Responses to Questions

**Comments to the Author**

1. If the authors have adequately addressed your comments raised in a previous round of review and you feel that this manuscript is now acceptable for publication, you may indicate that here to bypass the “Comments to the Author” section, enter your conflict of interest statement in the “Confidential to Editor” section, and submit your "Accept" recommendation.

Reviewer #1: All comments have been addressed

Reviewer #2: All comments have been addressed

2. Is the manuscript technically sound, and do the data support the conclusions?

Reviewer #1: Partly

Reviewer #2: Yes

3. Has the statistical analysis been performed appropriately and rigorously? 

Reviewer #1: No

Reviewer #2: Yes

4. Have the authors made all data underlying the findings in their manuscript fully available?

Reviewer #1: Yes

Reviewer #2: Yes

5. Is the manuscript presented in an intelligible fashion and written in standard English?

Reviewer #1: Yes

Reviewer #2: Yes

6. Review Comments to the Author

Reviewer #1: Thank you for the opportunity to review this paper again.

I believe that the descriptive statistics are of interest, but I still have some concerns about the analysis methods implemented in this study.

- The authors state that their hypotheses were generated post hoc, with no apparent hypotheses prior to starting the study. It is unclear, then, how much the findings reported here are subject to Type I error.

- As the hypotheses were data driven, statements in the discussion like “the first hypothesis… was supported by the data” is a little dishonest, as this then appears to have been an a-priori hypothesis.

- Other drug users were excluded, including those using armodafinil which is often used interchangeably, including as discussed by users in online forums

- It would be good in the supplementary documentation to have a copy of the survey and examples of how it was advertised to these forums

- It would be good to know specifically which subreddits were targeted (in supplementary material)

- The “design” section at the top of the methods still describes statistical methods, not the study design, which is better described under “Respondents” the “Design” section should say something about online survey etc. not the between group factors (which is still statistical methods, not design)

- In the calculation of Cohen’s d, which SD was used, was it from this or a reference population?

- The count data used in this study was probably not normally distributed as it was count data, not truly continuous, yet was treated like normally distributed data. Can the authors confirm if the data was somewhat normally distributed?

Reviewer #2: I appreciate the author's effort to revise the manuscript. As a psychologist, I strongly encourage the authors to continue this kind of work.

7. PLOS authors have the option to publish the peer review history of their article (what does this mean?). If published, this will include your full peer review and any attached files.

Reviewer #1: No

Reviewer #2: No

---

## [Author Response · Author response to Decision Letter 1]

12 Dec 2019

Dear Dr Hashimoto

Re. Manuscript ID PONE-D-19-19392R1 entitled “The off-prescription use of modafinil: An online survey of perceived risks and benefits”

Thank you for your email on the 1st November detailing the reviewers’ comments on the resubmission. We understand that, although you feel the manuscript has merit, it does not fully meet PLOS ONE’s publication criteria as it stands. We have carefully read through the points raised by reviewer one and have addressed each point below.

Reviewer #1: Thank you for the opportunity to review this paper again.

I believe that the descriptive statistics are of interest, but I still have some concerns about the analysis methods implemented in this study.

- The authors state that their hypotheses were generated post hoc, with no apparent hypotheses prior to starting the study. It is unclear, then, how much the findings reported here are subject to Type I error.

We thank the reviewer for raising these concerns. The aim of this study was to explore if there are any associations between patterns of use and perceived effects of modafinil outside of the laboratory. This is why the questions were phrased in the way they were. As no previous research has focused on this more ‘real-world’ experience of acute effects of modafinil, it was not possible to generate specific, directional hypotheses prior to the start of data collection. Once data had been collected we decided that the size of the sample and range of responses to the ‘number of perceived effects’ questions were best analysed using a 5 x 2 x 2 ANOVA. The specific hypotheses that are in this paper were generated based on the selection of this inferential test. 

- As the hypotheses were data driven, statements in the discussion like “the first hypothesis… was supported by the data” is a little dishonest, as this then appears to have been an a-priori hypothesis.

We appreciate the reviewer’s point. We did not intentionally mean to mislead and have changed the wording relating to the hypotheses where necessary. 

- Other drug users were excluded, including those using armodafinil which is often used interchangeably, including as discussed by users in online forums

We thank the reviewer for raising this point. We did not exclude people for taking armodafinil. Exclusions were for not taking modafinil as a cognitive enhancer (e.g. people who reported taking MPH) and for taking cognitive enhancers under prescription. For armodafinil in particular, none of our sample reported taking this when asked what other drugs they use alongside modafinil. A full list of the drugs mentioned in response to this question is now available as a supplementary table.

- It would be good in the supplementary documentation to have a copy of the survey and examples of how it was advertised to these forums

We appreciate the reviewer’s point. These have now been added to the supplementary materials (S1 and S3).

- It would be good to know specifically which subreddits were targeted (in supplementary material)

We, again, appreciate the reviewer’s point. This has now been added to the supplementary materials (S2)

- The “design” section at the top of the methods still describes statistical methods, not the study design, which is better described under “Respondents” the “Design” section should say something about online survey etc. not the between group factors (which is still statistical methods, not design)

The ‘design’ section has now been removed and, although we considered moving the contents of this section to the ‘respondents’ section, we felt that this fits better into the analysis section and we have now moved it to the start of the analysis section.

- In the calculation of Cohen’s d, which SD was used, was it from this or a reference population?

We thank the reviewer for asking for this clarification. The following sentence has now been added to the manuscript.

“Cohen’s d was calculated using the mean difference between the groups and dividing this by the pooled standard deviation.”

- The count data used in this study was probably not normally distributed as it was count data, not truly continuous, yet was treated like normally distributed data. Can the authors confirm if the data was somewhat normally distributed?

We thank the reviewer, this is a very good point. We have now checked for normality and have log-transformed the data and re-run the analyses. These show the same pattern of effects and we have provided these analyses in tabular form in the supporting materials. As log-transformed data are not so easy for the reader to interpret we have chosen to retain the original analyses in the write-up. However, if the Editor would prefer the log-transformed analyses we are happy to make this amendment.

Reviewer #2: I appreciate the author's effort to revise the manuscript. As a psychologist, I strongly encourage the authors to continue this kind of work.

We thank the reviewer for this kind encouragement.

---

## [Decision Letter · Decision Letter 2]

31 Dec 2019

The off-prescription use of modafinil: An online survey of perceived risks and benefits

PONE-D-19-19392R2

Dear Dr. Teodorini,

We are pleased to inform you that your manuscript has been judged scientifically suitable for publication and will be formally accepted for publication once it complies with all outstanding technical requirements.

With kind regards,

Kenji Hashimoto, PhD

Section Editor

PLOS ONE

Additional Editor Comments (optional):

Reviewers' comments:

Reviewer's Responses to Questions

**Comments to the Author**

1. If the authors have adequately addressed your comments raised in a previous round of review and you feel that this manuscript is now acceptable for publication, you may indicate that here to bypass the “Comments to the Author” section, enter your conflict of interest statement in the “Confidential to Editor” section, and submit your "Accept" recommendation.

Reviewer #2: All comments have been addressed

2. Is the manuscript technically sound, and do the data support the conclusions?

Reviewer #2: Yes

3. Has the statistical analysis been performed appropriately and rigorously? 

Reviewer #2: Yes

4. Have the authors made all data underlying the findings in their manuscript fully available?

Reviewer #2: Yes

5. Is the manuscript presented in an intelligible fashion and written in standard English?

Reviewer #2: Yes

6. Review Comments to the Author

Reviewer #2: The results described here are valuable information for considering promoting mental health of college students. I agree to the authors' conclusion that we should have a potential concern over the perception of modafinil as a "safe drug".

7. PLOS authors have the option to publish the peer review history of their article (what does this mean?). If published, this will include your full peer review and any attached files.

Reviewer #2: Yes: Naoyuki Hironaka

---

## [Editor Report · Acceptance letter]

14 Jan 2020

PONE-D-19-19392R2 

The off-prescription use of modafinil: An online survey of perceived risks and benefits 

Dear Dr. Teodorini:

I am pleased to inform you that your manuscript has been deemed suitable for publication in PLOS ONE. Congratulations! Your manuscript is now with our production department. 

With kind regards,

on behalf of

Prof. Kenji Hashimoto 

Section Editor

PLOS ONE